# Analysis of the Mechanical Characteristics of Human Pancreas through Indentation: Preliminary In Vitro Results on Surgical Samples

**DOI:** 10.3390/biomedicines12010091

**Published:** 2024-01-01

**Authors:** Michele Pagnanelli, Francesco De Gaetano, Arianna Callera, Gennaro Nappo, Giovanni Capretti, Silvia Carrara, Alice Maria Renata Ferrari, Francesco Cellesi, Maria Laura Costantino, Alessandro Zerbi

**Affiliations:** 1Department of Biomedical Sciences, Humanitas University, Via Rita Levi Montalcini 4, Pieve Emanuele, 20072 Milan, Italy; gennaro.nappo@hunimed.eu (G.N.); giovanni.capretti@hunimed.eu (G.C.); alessandro.zerbi@hunimed.eu (A.Z.); 2Pancreatic Unit, IRCCS Humanitas Research Hospital, Via Manzoni 56, Rozzano, 20089 Milan, Italy; francesco.degaetano@polimi.it (F.D.G.); arianna.callera@polimi.it (A.C.); 3Department of Chemistry, Materials and Chemical Engineering “Giulio Natta”, Politecnico di Milano, 20133 Milano, Italy; alicemaria.ferrari@polimi.it (A.M.R.F.); francesco.cellesi@polimi.it (F.C.); marialaura.costantino@polimi.it (M.L.C.); 4Endoscopy Unit, Department of Gastroenterology, IRCCS Humanitas Research Hospital, Via Manzoni 56, Rozzano, 20089 Milan, Italy; silvia.carrara@humanitas.it

**Keywords:** pancreaticoduodenectomy, learning curve, pancreatic consistency, phantom, indentation, elastic modulus

## Abstract

Pancreatic surgery is extremely challenging and demands an extended learning curve to be executed with a low incidence of post-operative complications. The soft consistency of the human pancreas poses a primary challenge for pancreatic surgeons. This study aimed to analyze the preliminary mechanical characteristics of the human pancreas to develop a realistic synthetic phantom for surgical simulations in the near future. Pancreatic specimens, comprehensive of the pancreatic parenchyma and main pancreatic duct, were collected during pancreatic resections and analyzed through nano-bio-indentation (Bioindenter^TM^ UNHT^3^ Bio, Anton Paar GmbH, Graz, Austria) to measure the elastic modulus. Comparisons were made between slow and fast loading rates, immediate and post-freezing analyses, and multipoint indentations. The results demonstrated that a slow loading rate (30 μN/min), immediate analysis, and multipoint measurements are crucial for obtaining accurate values of the elastic modulus of the human pancreas (1.40 ± 0.47 kPa). In particular, the study revealed that analysis after freezing could impact the outcomes of the indentation. Moreover, the study suggested that both the pancreatic parenchyma and the main pancreatic duct should be analyzed to achieve a more precise and comprehensive definition of the. mechanical features of the pancreas. These preliminary findings represent the initial steps toward defining the consistency and mechanical characteristics of human pancreatic tissue with the goal of creating a realistic synthetic phantom.

## 1. Introduction

Pancreatic surgery is technically challenging and not free from potential complications. A post-operative pancreatic fistula (POPF) stands out as the most prevalent and challenging complication, affecting up to 33% of patients, even in high-volume specialized centers [1,2,3]. High-grade POPF can give rise to other severe complications, such as delayed gastric emptying [4], post-pancreatectomy hemorrhage [5], sepsis, and, in extreme cases, death [6]. Consequently, these occurrences may result in prolonged hospital stays, swift clinical readmissions, and significant delays in oncological treatments, with notable repercussions on patient lifestyle, survival, and hospitalization costs [7,8]. However, no surgical techniques or patient management approaches have proven effective in reducing the incidence of POPF [9,10,11,12,13]. Conversely, it is widely acknowledged that the risk of developing POPF appears to be linked to the dimensions of the main pancreatic duct (MPD) and the consistency of the pancreatic parenchyma [14,15].

Since there is a limited number of studies exploring the mechanical characteristics of pancreatic tissue [16,17,18], this analysis relies on the subjective tactile sensation of the surgeon during pancreatic interventions [19]. In contrast, leveraging the collaboration between surgery and engineering, the present study aims to assess the mechanical properties of pancreatic parenchyma in terms of elasticity and response to deformation. These findings could facilitate the identification of a synthetic material with mechanical characteristics akin to the human pancreas. Such material could be instrumental in familiarizing surgeons with pancreatic anastomoses and developing innovative techniques with the goal of diminishing the incidence of POPF.

## 2. Materials and Methods

This study is the outcome of a collaboration among Humanitas University, Istituto Clinico Humanitas, and the Department of Chemistry, Materials, and Chemical Engineering “Giulio Natta” of Politecnico di Milano, aimed at fostering scientific innovation in pancreatic surgery.

In this research, we employed a nano-indentation tester designed for soft and biological materials (Bioindenter^TM^ UNHT^3^ Bio; Anton Paar GmbH, Anton Paar Graz, Austria) to estimate the elastic modulus (E) of the pancreatic parenchyma and the main pancreatic duct (MPD), with the goal of analyzing the mechanical characteristics of the human pancreas.

Between January and May 2023, we collected ten human pancreatic specimens in the operating room from patients undergoing pancreatoduodenectomy at the Pancreatic Surgery Unit of Istituto Clinico Humanitas. These samples were collected after the histopathological exclusion of tumor cells on the cut edge of the resected pancreas and were transported to a dedicated laboratory adjacent to the hospital for proper management. All specimens underwent immediate analysis and six of them were subsequently frozen at −80 °C and analyzed after defrosting. All procedures adhered to the ethical standards of the Institutional and National Research Committee, as well as the 1964 Helsinki Declaration and its later amendments or equivalent ethical standards. The study received approval from the Ethics Committee of Istituto Clinico Humanitas on 23 March 2022 (Protocol Code: RF-2021-12372635) and all patients provided written consent for all research procedures.

Upon arrival in the laboratory, the fresh pancreatic tissue was examined for dimensions and the linearity of the surface to be indented was assessed. The MPD was extracted from the sample and longitudinally cut to facilitate the indentation of the inner surface of the duct (Figure 1).

The pieces were subsequently glued to the Petri dish with cyanoacrylate to prevent any displacement during the indentation (Figure 2A,B). This step was crucial for accurately pinpointing each analyzed point and for conducting multiple measurements in the same region. Following this, the specimen was submerged in saline solution to preserve the physiological hydration of the tissue (Figure 2C) and the Petri dish was positioned in the machine with its orientation duly marked. 

The machine automatically mitigated the influence of external vibrations that could potentially disrupt the analyses thanks to the active vibration isolation unit, the Accurion Halcyonics i4 system (Accurion GmbH, Goettingen, Germany). The coordinates of each indentation point were recorded to facilitate the repetition of analyses at the same location. Before conducting the indentation test itself (Figure 3), the height of the sample’s surface was determined and saved through the adjust depth offset (ADO) procedure. This step is crucial when dealing with exceptionally soft tissues, such as pancreatic tissue. Failing to perform this operation correctly may result in an inaccurate surface identification, causing the indentation to begin inside the sample or upon contact with the saline solution.

The parameters to set were the following:-Maximum load [µN]: load to be reached during the loading phase;-Loading rate [µN/min]: speed to reach the maximum load;-Pause [s]: time of maintenance of the maximum load;-Unloading rate [µN/min]: speed at the removal of the maximum load;-Approach speed [µm/min]: speed to approach to the specimen;-Retract speed [µm/min]: speed to separate from the specimen;-Contact stiffness threshold [µN/µm]: measure that enables the machine to recognize the contact with the specimen to analyze.

Once the indenter concluded its analyses, the results were presented in the form of (Figure 4):-Force–time and displacement–time curves: these curves allow for the identification of the approach, loading phase, pause, and unloading phase (Figure 4A);-Force–displacement curve: this curve enables the calculation of the E and illustrates the viscoelastic properties of the tissues (Figure 4B);-Hertz’s modulus (*E_Hz_*): this value estimates the E of the analyzed soft material.

The *E_Hz_*, also known as Hertz’s modulus, serves as a physical measure of tissue elasticity, based on Hertz’s contact stress theory [20]. This theory proves more suitable than the Oliver and Pharr approach when analyzing soft biological specimens using a rigid spherical probe. Specifically, *E_Hz_* represents a characteristic of an elastic material, arising from the deformation induced during loading and the subsequent recovery observed upon unloading. Typically calculated as the ratio of stress to strain along the same axis within the stress range governed by Hooke’s law, *E_Hz_*, in this context, is estimated from the force–displacement curve generated by the indenter at the conclusion of the procedure.

The *E_Hz_* is calculated from the reduced modulus (*E_r_*), the mechanical properties of the indenter, and the sample, as detailed in the following equations:(1)$$E_{Hz}=\frac{1-\nu_{s}^{2}}{\frac{1}{E_{r}}-\frac{1-\nu_{i}^{2}}{E_{i}}}$$
where $v_{s}\;$and $v_{i}$ represent the Poisson’s ratios of the specimen (set at 0.46) and of the indenter (set at 0.07), respectively, and *E_i_* is the elastic modulus of the indenter (set at 1141 GPa). The *E_r_*, a fictitious value in which the elastic deformation of the sample and the indenter are combined, is estimated from the indentation as follows:(2)$$E_{r}=\frac{3}{4}\cdot \frac{F_{n}}{\sqrt{R}\cdot {\left(h-h_{offset}\right)}^{\frac{3}{2}}}$$
where *F_n_* [N] is the indentation force, *R* [m] is the radius of the rigid spherical probe, *h* [m] is the depth of the indentation probe, and *h_offset_* [m] is the depth of the layers over the material.

The Nano-bioindenter utilized in this study features a ruby spherical probe with a diameter of 1000 μm; the E and Poisson’s ratios were provided by the manufacturer. The indentation procedure involves generating a distinctive force–displacement curve characteristic of the analyzed tissues.

The force and displacement versus time curves obtained during the indentation exhibit distinct phases:-Approach: during this phase, the indenter moves towards the sample without making contact;-Load: when the probe makes contact with the tissue, the load phase commences, reaching the previously set maximum load. This is followed by a constant load period;-Unload: this phase marks the conclusion of the test.

Given the viscoelastic and highly soft nature of the human pancreas, obtaining reliable force–displacement curves requires the following:-Low contact stiffness threshold (0.1–0.2 µN/µm): a low contact stiffness is imperative for accurately identifying the surface of the soft pancreatic tissue;-Slow loading rate (30 µN/min): viscoelastic materials naturally exhibit rate dependence, requiring the loading to approximate a quasi-static condition to obtain reliable values of E.

In summary, good force–displacement curves should exhibit the following:-A clean approach phase;-Absence of oscillations or irregularities;-Consistency between the peak of the curve and the maximum load;-Correct closure at the conclusion of the analyses.

We opted to analyze some of the samples after freezing to assess whether the mechanical characteristics of human pancreatic tissue remain consistent. Freezing, unlike cryopreservation, not only preserves cell vitality but also maintains the mechanical properties of the analyzed tissues. Previous studies on soft materials have demonstrated that viscoelastic features remain unchanged after freezing at −80 °C for 14 days, showing no statistically significant differences [21,22,23]. Nevertheless, due to the lack of similar tests involving pancreatic tissue, we chose to compare fresh samples with those post-freezing, analyzing the same sample at identical points before and after freezing. Building on previous studies conducted on various biological tissues [23], we adopted a gradual defrosting protocol. All the samples were kept at −20 °C for 12 h and subsequently at 5 °C for the next 5 h. Following this, they were hydrated with saline solution for 1 h and subjected to indentation once they reached a stable temperature of 18–20 °C.

For each sample, a range of 3 to 5 points was tested, depending on the size of the specimen. These points were distributed as evenly as possible across the sample, maintaining a minimum distance of at least 2 mm to prevent any overlapping of the indentation areas. Points featuring irregular surfaces, such as peaks and valleys, were intentionally avoided to ensure the generation of clean indentation curves. Table 1 provides details on the number of samples and points considered in each analysis.

For statistical significance, five indentations were carried out at each point with 10-min intervals. The inter-indentation pause was deemed necessary due to the absence of cyclic characterization for pancreatic tissue, which makes it challenging to predict the impact of multiple consecutive indentations. A 10-min interval was found to be sufficient, as no discernible trend, whether in terms of hardening or softening, was evident in our tests.

All indentations were conducted with a maximum load of 30 µN, which proved to be the optimal value for obtaining well-defined indentation curves while avoiding excessive depth. Various loading times were explored to analyze the rate effect on the material and determine the most suitable total loading time for indentation. This consideration was crucial as it influences the mechanical response of the tissue and, consequently, the obtained *E_Hz_* value. The initial loading time considered was 10 s, as suggested in the Anton Paar manual for soft tissue indentation. While no specific standards exist for soft material indentation, the authors attempted to align with the rationale behind other standards for viscoelastic materials, suggesting low velocities to obtain quasi-static tests [24]. The goal was to decrease the velocity progressively until no further changes were observed, resulting in an increase to 60 s and then to 120 s. With a maximum load of 30 µN, these total loading times corresponded to loading rates of 180, 30, and 15 µN/min, respectively.

The pause during indentation was set at 5 s to ensure that the maximum load was reached before the unloading step commenced. The unloading was executed at the same speed as the loading.

### Statistical Analysis

We conducted an analysis and comparison of the mean values of E_Hz_ to identify the presence of statistically significant differences during multipoint indentations performed at various loading rates and before and after freezing the samples.

The Shapiro–Wilk test was employed to assess the normality of the groups involved in the analyses. To examine statistically significant differences between fresh and frozen samples, a paired Student’s *t*-test or Wilcoxon signed-rank test, depending on the normality of the considered groups, was performed.

For other tests, we utilized one-way analysis of variance (ANOVA) for data with a normal distribution, while for data without a normal distribution, Kruskal–Wallis tests were applied. Post-hoc tests were carried out to identify statistically significant differences in our analysis. *p* values < 0.05 were considered statistically significant.

Statistical analysis was conducted using SPSS software (SPSS Inc., version 27 for Macintosh, IBM, Chicago, IL, USA).

## 3. Results

### 3.1. Loading Rate

The mechanical characteristics of viscoelastic materials are influenced by the speed of the applied stimulation, determined by the maximum load of indentation and the total loading time. A comparison was made among loading rates of 180, 30, and 15 µN/min on two samples, while for the comparison between 30 and 15 µN/min, two additional samples were included. Multiple points in each sample were tested to ensure more accurate results.

In the following section, each indentation point is identified by a combination of the sample number (1, 2, 3, and 4) and a letter (A, B, and C) to distinguish different positions within a sample. Points A and C were located at the extremities of the sample (while maintaining a sufficient distance from the margin to avoid edge effects), while point B was in the central area. Table 2 provides details on all the analyzed points.

The analysis of two samples, one of which is shown in Figure 5, revealed a statistically significant difference between the *E_Hz_* obtained with slow loading (180 µN/min) and the one with a fast loading rate (30 µN/min). In particular, this resulted in a considerably lower E_Hz_ with slow loading due to the progressive adjustment of the viscoelastic tissue to the compression of the indenter.

Regarding the differences between slow (30 µN/min) and extra-slow (15 µN/min) loading rates, the analysis performed after averaging all the samples revealed no statistically significant differences with an average of 1.98 ± 0.92 for the slow and of 1.64 ± 0.61 for the extra slow loading rates. When the samples were considered separately (Figure 6), the differences were revealed to be statistically significant only in the third analysis.

The appropriate loading rate for the accurate analysis of pancreatic tissue was established to be 30 µN/min, obtained by combining a maximum load of 30 µN and a loading time of 60 s. This decision took into account the viscoelastic nature of the pancreas and the stability of the test. Viscoelastic tissues tend to be adversely affected by fast loading rates (e.g., 180 µN/min) as this induces greater resistance of the tissue to the compression of the indenter, resulting in higher *E_Hz_* values. Conversely, slower loading rates allow viscoelastic materials to adapt to the compression of the indenter, which experiences lower resistance and yields more accurate E_Hz_ values. However, with an extra-slow loading rate (15 µN/min), some indentations were not executed correctly, as indicated by indentation curves with more artifacts and occasional saturation of the displacement range. Consequently, the selected rate was a compromise between the desired slow rate and the limitations encountered at lower rates, which prevented the attainment of consistent results.

### 3.2. Sample’s Homogeneity

The pancreatic parenchyma exhibits extensive heterogeneity due to its fibrotic and glandular nature. This heterogeneity is evident not only across different samples but also within various areas of the same sample. Consequently, to achieve a more precise analysis of pancreatic specimens, mechanical properties were measured using a multipoint indentation technique. This approach allowed the mean value of the E_Hz_ for each anatomical piece to be obtained. Each specimen underwent analysis at five different points, whenever possible, and the measurement at each point was repeated five times, as previously explained. Force–displacement curves were generated (see Figure 7) and a mean value of EHz, along with the confidence interval, was calculated for each point (refer to Table 3).

Based on the statistical tests conducted, the five points were not found to be equivalent. Specifically, point A showed statistically significant differences from all others, except for D. Moreover, several other differences emerged between the points. The detailed comparisons between each point are presented in Table 4. Considering the observed statistically significant differences within the same sample, it was concluded that a minimum of three points needed to be analyzed for each sample to ensure a correct characterization.

### 3.3. Freezing

Six of the samples, analyzed immediately after the intervention, underwent freezing at −80 °C for a maximum of 30 days in order to compare the results of indentations before and after the freezing step. The Petri dish was appropriately marked during the initial test, allowing each specimen to be indented at the same points before and after the freezing process (refer to Figure 8, Table 5 and Table 6).

Contrary to findings in the literature for soft tissues [25], the results indicated a statistically significant difference in the average E_Hz_ of the pancreatic parenchyma before (2.12 ± 0.90 kPa) and after (1.86 ± 0.87 kPa) the freezing process.

According to these findings, we can conclude that the pancreatic tissue should be characterized when fresh since the freezing process could significantly alter the mechanical properties of the tissue.

### 3.4. Main Pancreatic Duct

The portion of the specimen including the main pancreatic duct (MPD) was isolated and longitudinally cut to enable indentation of the inner surface of the duct. Each duct underwent indentation at three to five points, depending on the size of the sample. At each point, the indentation was repeated five times at 10-min intervals. The results are illustrated in Figure 9.

Given its histological nature, it was anticipated that the MPD would be more rigid than the surrounding acinar parenchyma. However, our preliminary tests revealed that the MPD has an indentation modulus very similar to the pancreatic tissue (*E_Hz_* of the MPD: 1.71 ± 0.88 kPa versus E_Hz_ of the pancreatic parenchyma: 1.69 ± 0.79 kPa). It is noteworthy that in sample 3, the exceptionally high modulus of the duct corresponds to an unusually elevated *E_Hz_* of the pancreatic parenchyma (average *E_Hz_*: 2.36 kPa). These findings require further validation and confirmation through additional analyses of other samples.

## 4. Discussion

Pancreatic surgery stands as one of the most challenging surgical fields, requiring an extensive learning curve to achieve a low incidence of post-operative complications [26,27,28,29]. While post-operative mortality has gradually decreased in high-volume surgical centers, the morbidity rate following pancreatic resections remains relatively stable [30].

Of all the complications arising after pancreatoduodenectomy, POPF is the most common and can either remain isolated or become the root cause of additional post-operative issues. The soft consistency of the pancreas and the small dimensions of the MPD are well-known major contributors to POPF [14,15], and as of now, no surgical techniques have proven successful in reducing its incidence [9,10,11,12,13]. However, the creation of a pancreatic phantom composed of specific tissue with mechanical characteristics. similar to the human pancreas could prove highly beneficial. Such a phantom would enable pancreatic surgeons to hone their surgical skills on a structure possessing analogous properties to what they encounter in the operating room.

In the literature, few studies have examined the consistency of pancreatic parenchyma. Older techniques relied on shear rheology [31,32], while more recent ones employ the indentation technique [16,33].

Nicolle et al. conducted an analysis of the viscoelastic properties of the porcine pancreas through rheological techniques, comparing it with other organs (kidney, liver, and spleen). Their findings concluded that pancreatic parenchyma exhibits the highest viscosity [31]. Similar results were observed in a study by Wex et al., where human and porcine pancreases were analyzed using shear rheology. They concluded that the pancreas displays a nonlinear viscoelastic nature due to its fibrous connective structure associated with acinar cells and islets of Langerhans [32]. However, rheology assesses tissues as a whole without the ability to perform point-specific analyses or conduct multiple assessments on a single specimen. In contrast, recent techniques for analyzing biological tissues, such as indentation, measure the consistency of specific areas within tissues and can be repeated on the same sample.

Li et al. presented a prototype of a portable pen-sized indenter to measure the elastic modulus (*E_Hz_*) of soft tissues in vivo, demonstrating results from tests on mouse pancreases [33]. While this innovative instrument facilitates the measurement of organ consistency without the need for surgical resections, conducting such measurements in the operating room without isolated support could yield altered results influenced by environmental factors. The E_Hz_ values obtained with this instrument generally tend to be lower than those observed in our study or in other works on the same topic (0.41–0.78 kPa vs. 0.84–4.08 kPa). However, this could also be influenced by the fact that Li et al. analysis was limited to mouse pancreas, which may have different mechanical characteristics from the human pancreas.

Conversely, Sugimoto et al., through indentation of human pancreatic specimens, showed *E_Hz_* values much more similar to those obtained in our study [16]. In their work, the Japanese team also established a correlation between E_Hz_ measured through indentation, the tactile evaluation of the surgeon, the anatomopathological level of fibrosis, and the development of POPF. However, although Sugimoto et al. performed multipoint indentations of the pancreatic parenchyma, they repeated the test three times at each point without pauses, potentially impacting the accuracy of the average results. Additionally, they did not measure the mechanical characteristics of the MPD, which could provide important information about the fibrotic and ductal components of the pancreatic gland.

In contrast, we argue that for a more precise and comprehensive understanding of the mechanical characteristics of the human pancreas, encompassing all its components, both the pancreatic parenchyma and MPD should undergo analysis. This study, thanks to a collaborative effort between surgery and engineering, measured the consistency of the human pancreas with the aim of identifying a synthetic material to create a realistic phantom. Such a phantom could serve as a valuable tool for surgical training and as a scaffold for future innovations. This ambitious initiative would empower pancreatic surgeons to practice pancreatic anastomosis using various surgical techniques and different suture threads on a phantom with viscoelastic features realistically comparable to the human pancreas. In the existing literature, Zhang et al., in their meta-analysis, compared pancreaticogastrostomy and pancreaticojejunostomy reconstruction after pancreaticoduodenectomy [34], while Pagnanelli et al. conducted in vitro analyses of the mechanical characteristics of different suture threads used for pancreatic anastomosis after exposure to bile and pancreatic juice [35]. With the availability of a realistic pancreatic phantom, different anastomotic techniques could be evaluated and the use of different surgical threads could be repeatedly experimented with in a risk-free environment.

This study is primarily focused on identifying a precise testing protocol that would facilitate the comprehensive mechanical characterization of pancreatic tissue and the identification of a corresponding artificial material.

Concerning the analysis of parenchymal tissue, given its viscoelastic nature, we demonstrated the significance of performing tests under quasi-static conditions and allowing ample time for material recovery between tests. In this study, the appropriate rate identified for measurements was 30 μN/min, although additional tests should be conducted to analyze the cyclic behavior of the tissue and determine the optimal waiting time for recovery and testing optimization. Furthermore, we established that when conducting local analysis on the pancreas, multiple points should be tested to gather comprehensive information on the specimen. The number of positions should be determined based on the size of the sample without overlapping the indented areas. Lastly, according to the findings of this paper, pancreatic tissue should always be analyzed when fresh to avoid any alteration of the mechanical study. However, we do not rule out the possibility that a different freezing and thawing protocol could be effective in preserving the mechanical properties of the tissue; we plan to further investigate this issue.

Concerning the analysis of the MPD, additional measurements are necessary to obtain more precise and comprehensive results. It is essential to completely isolate the duct from the parenchyma to rule out any potential influence of the underlying tissue layer on the resulting modulus. Additionally, the tensile properties of the duct should be analyzed. Given the histological structure of the duct, predominantly composed of connective tissue, its compression properties may be similar to those of the pancreatic parenchyma, while its tensile characteristics, linked to tear resistance, could be entirely different. Therefore, it is crucial to conduct a viscoelastic characterization of both the pancreatic parenchyma and the MPD. This analysis holds fundamental importance in the creation of a more realistic pancreatic phantom capable of simulating the surgical conditions faced by pancreatic surgeons in performing pancreatic anastomosis.

Nevertheless, this study has some limitations. Initially, due to the necessity of accurately calibrating the indenter function, the number of patients analyzed thus far is relatively small. However, we have been collecting pancreatic specimens from all patients undergoing pancreatoduodenectomy or total pancreatectomy in our center, and new data will soon be available for a more precise definition of the mean E_Hz_ of the human pancreas. Second, in these tests, the indentation parameters were often varied. Nonetheless, we have now successfully fine-tuned these parameters and subsequent tests will be conducted under standardized conditions. Third, the present data lack correlation with endoscopic and anatomopathological information. As part of this project, all patients entering the study will undergo preoperative evaluation with endoscopic elastography to measure the stiffness of healthy and pathological parenchyma; all pancreatic specimens will be microscopically analyzed to measure the level of fibrosis. An additional scope of this study will be the correlation between the numerical stiffness scale obtained with indentation and the qualitative information commonly obtained during clinical exams.

## 5. Conclusions

Numerous innovations have been introduced in pancreatic surgery but it remains a challenging field that demands a prolonged learning curve to enhance surgical techniques and achieve low levels of post-operative complications. The inherent soft and viscoelastic nature of the human pancreas, attributed to its fibrotic and glandular structure, poses a significant challenge for pancreatic surgeons.

The creation of an artificial surgical model mirroring the mechanical characteristics of the human pancreas would facilitate the simulation of surgical procedures, enabling surgeons to gain confidence in dealing with pancreatic consistency. A realistic pancreatic phantom could be valuable for evaluating different anastomotic techniques and experimenting with various surgical threads in a risk-free environment.

This study, arising from a collaborative effort between surgery and engineering, presents preliminary findings on the mechanical features of the human pancreas using nano-bio-indentation. This work represents a crucial initial step, as the estimated modulus serves as the primary criterion for selecting artificial materials. Only materials exhibiting a matching indentation modulus will undergo further analysis of their viscoelastic characteristics for the fabrication of the pancreatic phantom. Once the material is selected, a 3D model of the organ will be reconstructed from clinical images, modified to exclude irrelevant tissues, and fabricated using 3D printing or casting techniques.

Based on our findings, pancreatic specimens should be analyzed promptly after collection from the operating room, as freezing may lead to altered or irregular outcomes during indentation. Moreover, considering the viscoelastic nature of the human pancreas, multipoint indentations with slow loading are preferable for more precise results in terms of E_Hz_. Additionally, both pancreatic parenchyma and the MPD should undergo analysis.

With these pivotal preliminary results, the present work establishes the groundwork for realizing a realistic synthetic phantom of the human pancreas, poised to be utilized in various surgical and medical procedures.

## Figures and Tables

**Figure 1 biomedicines-12-00091-f001:**
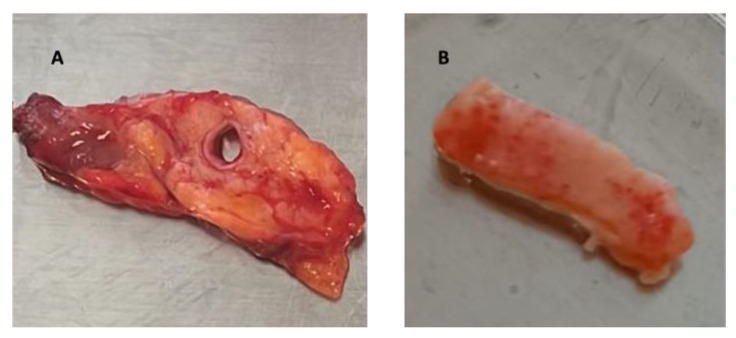
MPD before (**A**) and after (**B**) being separated from the sample and cut for testing on the inner surface.

**Figure 2 biomedicines-12-00091-f002:**
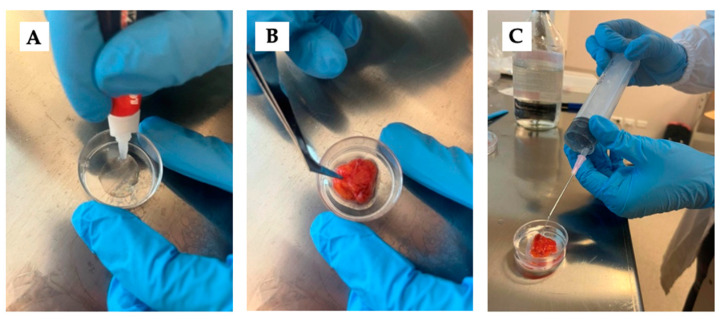
Sequence of sample preparation: the glue was placed on the Petri dish (**A**), the sample was placed on the glue with the smoother side exposed (**B**), and the sample was submerged in physiological solution after waiting a few minutes for the glue to dry (**C**).

**Figure 3 biomedicines-12-00091-f003:**
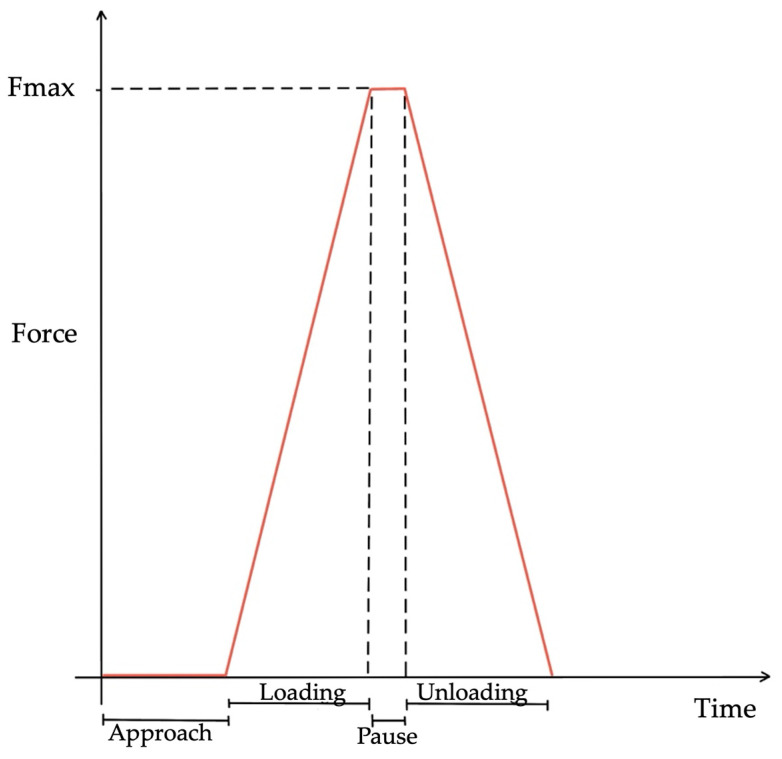
Indentation phases: Approach, the indenter moves towards the sample with a null force registered; Loading: it starts when contact happens between the indenter and the sample and ends when the maximum force is reached; Pause: the maximum force is held constant; Unloading: the indenter moves up until no more force is registered.

**Figure 4 biomedicines-12-00091-f004:**
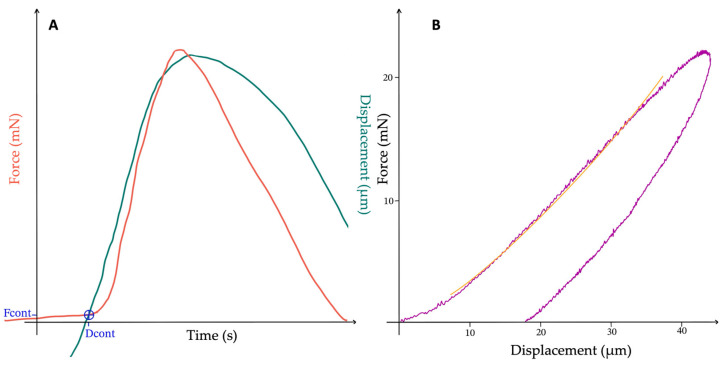
Curves generated by the indenter: force (red) and displacement (green) with contact point determination (blue) vs. time (**A**), force VS displacement (purple) with the *E_Hz_* estimation (orange) (**B**).

**Figure 5 biomedicines-12-00091-f005:**
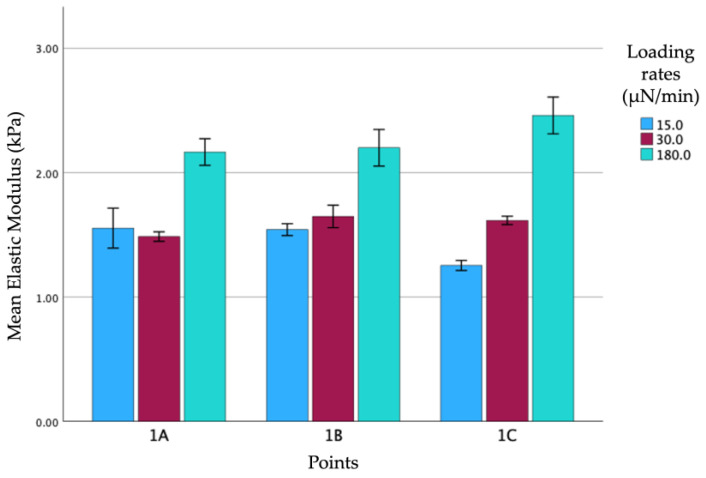
Comparison of different loading rates for three different points of the same specimen (**1A, 1B, 1C**): blue (**180 µN/min**), reddish-purple (**30 µN/min**), and light blue (**15 µN/min**). For each condition considered, the error is reported as the standard deviation.

**Figure 6 biomedicines-12-00091-f006:**
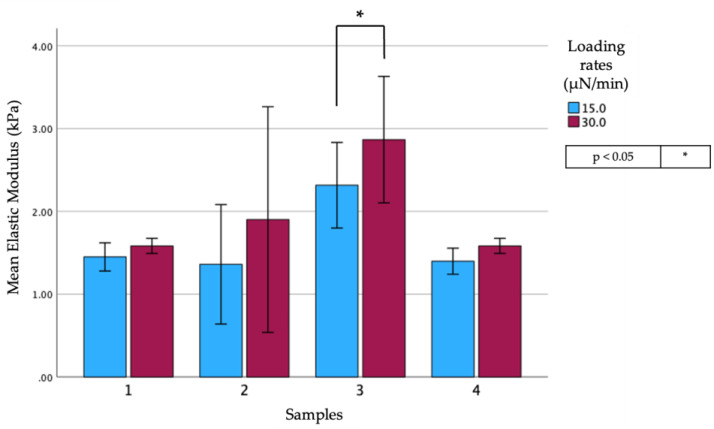
Comparison of the mean values of E_Hz_ of three different samples analyzed at two long loading times: reddish-purple (**30 µN/min**) and blue (**15 µN/min**).

**Figure 7 biomedicines-12-00091-f007:**
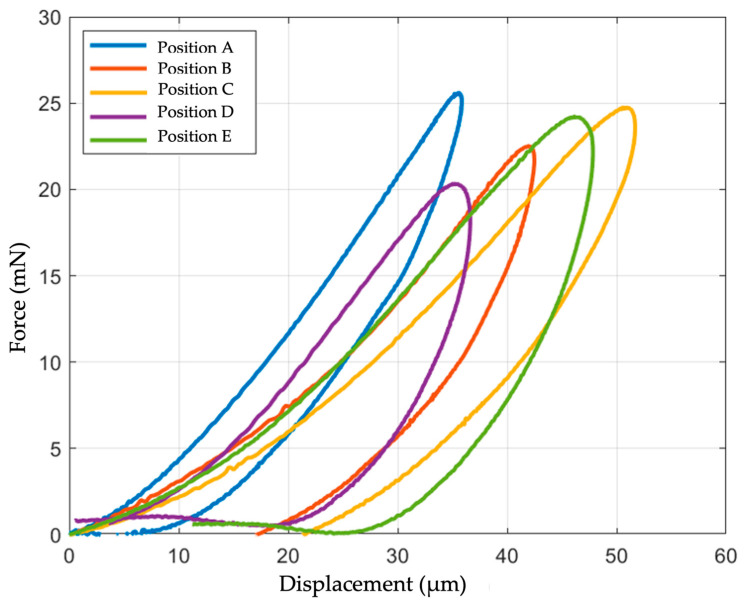
Stress–strain curves of multipoint indentation of pancreatic parenchyma on the same sample in five different positions.

**Figure 8 biomedicines-12-00091-f008:**
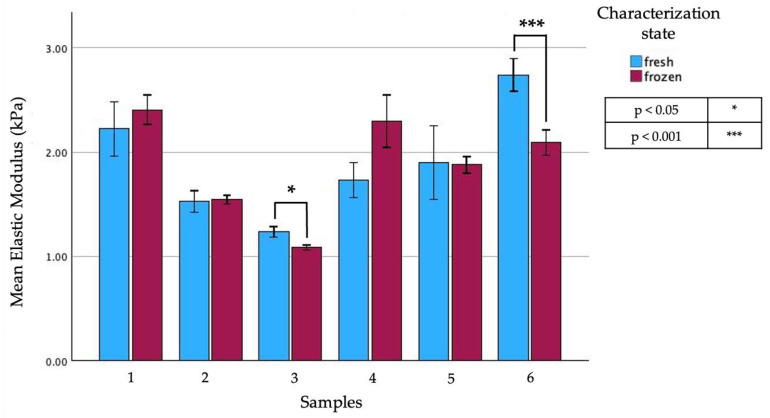
Comparison of the mean values of the *E_Hz_* of six different samples of pancreatic parenchyma before and after freeing.

**Figure 9 biomedicines-12-00091-f009:**
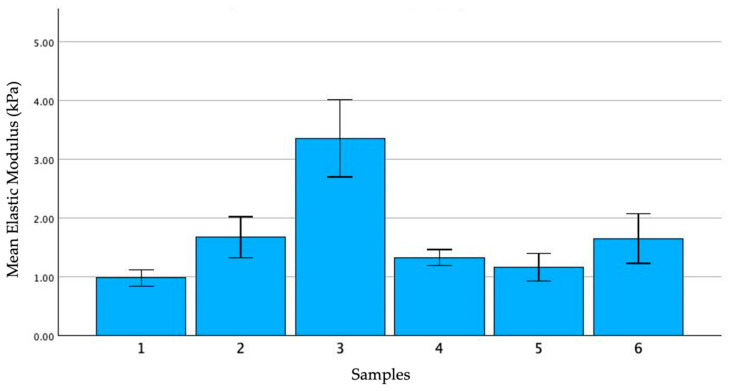
Comparison of the mean values of the *E_Hz_* in multipoint indentation of the MPD for different samples.

**Table 1 biomedicines-12-00091-t001:** Number of samples and points analyzed for each of the following sections.

Analysis	Samples	Total Points
Loading rate	4	12
Homogeneity	1	5
Freezing	6	21
Main pancreatic duct	6	23

**Table 2 biomedicines-12-00091-t002:** Mean *E_Hz_* [kPa] of three different areas of indentation (**A, B, and C**) with three different loading rates (**180 µN/min, 30 µN/min, and 15 µN/min**) on each of the four analyzed samples.

*E_Hz_* [kPa]			
Position	Loading Rate 180 µN/min	Loading Rate 30 µN/min	Loading Rate 15 µN/min
1A	2.17 ± 0.11	1.49 ± 0.04	1.55 ± 0.16
1B	2.2 ± 0.15	1.65 ± 0.09	1.54 ± 0.05
1C	2.46 ± 0.15	1.62 ± 0.03	1.25 ± 0.04
Mean 1	**2.17 ± 0.06**	**1.58 ± 0.09**	**1.45 ± 0.17**
2A	2.17 ± 0.10	3.61 ± 0.92	2.34 ± 0.09
2B	2.20 ± 0.15	1.26 ± 0.24	0.85 ± 0.02
2C	2.46 ± 0.15	0.84 ± 0.18	0.9 ± 0.14
Mean 2	**2.28 ± 0.18**	**1.9 ± 1.36**	**1.36 ± 0.72**
3A	-	2.26 ± 0.16	2.22 ± 0.15
3B	-	2.48 ± 0.29	1.83 ± 0.07
3C	-	3.85 ± 0.28	3.04 ± 0.08
Mean 3		**2.87 ± 0.76**	**2.32 ± 0.52**
4A	-	1.49 ± 0.04	-
4B	-	1.65 ± 0.09	1.54 ± 0.05
4C	-	1.62 ± 0.03	1.25 ± 0.4
Mean 4		**1.58 ± 0.09**	**1.40 ± 0.16**

**Table 3 biomedicines-12-00091-t003:** Comparison of the mean values of the *E_Hz_* in multipoint indentation of the pancreatic parenchyma of the same sample.

Position	*E_Hz_* [kPa]
A	3.24 ± 0.21
B	1.90 ± 0.12
C	1.98 ± 0.11
D	2.92 ± 0.34
E	2.13 ± 0.15

**Table 4 biomedicines-12-00091-t004:** Statistical significance between the positions analyzed in Table 3. Each row tests the null hypothesis that the Sample 1 and Sample 2 distributions are the same. The significance level is 0.05. ^a^ Significance values have been adjusted by the Bonferroni correction for multiple tests.

Statistical Significance		
Sample 1–Sample 2	Sig.	Adj. Sig ^a^
B–C	0.591	1.000
B–E	0.144	1.000
B–D	0.002	0.024
B–A	<0.001	0.002
C–E	0.355	1.000
C–D	0.013	0.126
C–A	0.002	0.017
E–D	0.117	1.000
E–A	0.027	0.268
D–A	0.519	1.000

**Table 5 biomedicines-12-00091-t005:** Comparison of the mean values of the *E_Hz_* of five different samples of pancreatic parenchyma before and after freeing.

*E_Hz_* [kPa]		
Sample	Before Freezing	After Freezing
**1**	2.22 ± 1.16	2.41 ± 0.63
**2**	1.53 ± 0.31	1. 54 ± 0.18
**3**	1.24 ± 0.19	1.09 ± 0.09
**4**	1.73 ± 0.74	2.30 ± 1.12
**5**	1.90 ± 1.36	1.88 ± 0.31
**6**	2.74 ± 0.70	2.09 ± 0.54

**Table 6 biomedicines-12-00091-t006:** Comparison of the comprehensive mean values of the EHz of pancreatic parenchyma before and after freeing.

Comprehensive *E_Hz_* [kPa]	
**Before Freezing**	2.12 ± 0.90
**After Freezing**	1.86 ± 0.87

## Data Availability

Data are unavailable due to privacy or ethical restrictions.

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
