# Peer review of "Analysis of the Mechanical Characteristics of Human Pancreas through Indentation: Preliminary In Vitro Results on Surgical Samples"

_biomedicines, 2024, doi:10.3390/biomedicines12010091_

Round 1
Reviewer 1 Report
Comments and Suggestions for Authors
This manuscript describes an interesting analysis of the mechanical characteristics of pancreatic parenchyma in terms of elasticity and response to deformation. Human pancreatic specimens were collected and studied in a nano-indentation tester to measure the elastic modulus in order to analyze the mechanical characteristics of pancreas. The topic is very relevant in the context of finding synthetic tissues with similar mechanical characteristics to human pancreas. In my opinion, the manuscript could be published after major revision:
a) In page 4, figure 3 A and B, the authors should include the title of both axis as well as the units of the variables described in the graph.
b) The authors should describe the meaning of the variable Er (reduced E)
c) In page 5, second paragraph, it is stated that “The Nano-bioindenter used in this study has a ruby spherical probe with a diameter of 1000 mm”. I think that may be an error in the dimension of the ruby spherical probe.
d) In page 6, figure 4 and figure 5, the authors should specify that the variable measured in the y-axis is the mean elastic modulus (kPa)
e) The authors should describe with more detail the meaning of the position 1A, 1B, and 1C, as well as 2A, 2B, and 2C.
f) In page 7 second paragraph it is stated that no statistically significant differences were found between the short and extra short loading rates. But Figure 5 and Table 2 suggest that there are significant differences between the two above mentioned loading rates in position 2A. Please comment on this issue.
g) The same thing can apply for the results in Figure 6 and Table 3. The results suggest that there are significant differences between mean elastic modulus for the short and extra short loading rates in position 3B.
h) In page 10 and page 11 there are two section with the same title “Homogenicity”. The authors should use a more descriptive title for these two sections.
i) In page 13, the section “4.Discussion” is focused on discussing the work reported in the literature. The authors should include in this section a discussion of their results, including the relevance of their findings, as well as the implications of their results.
Author Response
Dear Reviewer,
Thank you for your review of our paper. We have answered each of your points below.
- a) In page 4, figure 3 A and B, the authors should include the title of both axis as well as the units of the variables described in the graph.
Thank you for your comment, the figure has been updated in the hopes of increasing its clarity.
- b) The authors should describe the meaning of the variable Er (reduced E)
The equations were rewritten in order to improve readability of the manuscript, and a definition of the variable was included:
“The reduced modulus Er [Pa], a fictitious value in which the elastic deformation of the sample and the indenter are combined, is estimated from the indentation”
- c) In page 5, second paragraph, it is stated that “The Nano-bioindenter used in this study has a ruby spherical probe with a diameter of 1000 mm”. I think that may be an error in the dimension of the ruby spherical probe.
We apologise as that was a mistake on our part, the diameter has been changed to the correct value of 1000 μm.
- d) In page 6, figure 4 and figure 5, the authors should specify that the variable measured in the y-axis is the mean elastic modulus (kPa)
Following your suggestion all similar figures have been updated.
- e) The authors should describe with more detail the meaning of the position 1A, 1B, and 1C, as well as 2A, 2B, and 2C.
Following you suggestion this phrase was added in pg 6 in the hope of increasing the clarity of the manuscript:
“In the following section each indentation point is identified with the combination of the sample number (1, 2, 3, 4) and a letter (A, B, C) to distinguish the different positions within a sample. Points A and C were identified at the extremities of the sample (remaining far enough away from the margin to avoid edge effects), while point B in the central area”
- f) In page 7 second paragraph it is stated that no statistically significant differences were found between the short and extra short loading rates. But Figure 5 and Table 2 suggest that there are significant differences between the two above mentioned loading rates in position 2A. Please comment on this issue.
Thank you for your suggestion. Figures and tables concerning this topic have been updated in the hopes of increasing the clarity of the manuscript. Moreover, the explanation on pg 8 and 9 has been expanded.
- g) The same thing can apply for the results in Figure 6 and Table 3. The results suggest that there are significant differences between mean elastic modulus for the short and extra short loading rates in position 3B.
Thank you for your suggestion. As mentioned above, figures and tables concerning this topic have been updated in the hopes of increasing the clarity of the manuscript and the explanation on pg 8 and 9 has been expanded.
- h) In page 10 and page 11 there are two sections with the same title “Homogenicity”. The authors should use a more descriptive title for these two sections.
We apologise as that was a mistake on our part, the title of the section at page 11 has been changed to “Freezing”.
- i) In page 13, the section “4.Discussion” is focused on discussing the work reported in the literature. The authors should include in this section a discussion of their results, including the relevance of their findings, as well as the implications of their results.
Thank you for you comment. The mention section has been revised and discussion of the results significantly expanded. In particular the following paragraph has been added on pg 16:
“This work concerned itself with the identification of a precise testing protocol that will allow the complete mechanical characterization of the pancreatic tissue and the identification of a corresponding artificial material. Concerning the parenchymal tissue, first of all, dealing with a viscoelastic material it is of trivial importance to perform the tests in quasi static conditions and to leave the material time for recovery between tests. In this work the identified appropriate rate is 30 μN/min. Additional tests should be performed to analyze the cyclic behavior of the tissue and identify the appropriate waiting time to allow recovery and optimize testing. Secondly, we established that, when performing local analysis on the pancreas, several points should be tested in order to gather complete information on the specimen. The number of positions should be determined based on the size of the sample without having overlap of the indented areas. Lastly, according to the findings of this paper pancreatic tissue should always be analyzed fresh to avoid any alteration of the mechanical study. The authors do not exclude that a different freeze and thaw protocol could be effective in preserving the mechanical properties of the tissue and will further investigate the issue.”

Reviewer 2 Report
Comments and Suggestions for Authors
The authors characterized the mechanical properties of human pancreas parenchyma, using clinically isolated human samples that include both parenchyma and main pancreatic duct, towards providing data that would inform the creation of a realistic synthetic phantom for surgical simulations and training. Such phantom-based training can help eventually shorten the long learning curve associated with the complexity of pancreatic surgery and reduce the post-operative complications’ frequency.
However, this study only represents initial characterization comparing resected pancreatic samples using different characterization protocols (slow vs. fast, single vs. numerous measurement points etc.) and sample prep (immediately after harvesting vs. after a freeze and thaw cycle). Overall, the work suffers from many major and some minor issues, and cannot be accepted for publication in its current form without substantial revisions that include improved and additional data informative presentation, improved English, and making sure the manuscript is coherent and emphasizes the immediate (not theoretical) significance of their findings over the current available literature.
1. Major comments:
a. Overall the work presented is not sufficiently coherent. To name but a few examples:
i. The optimal rate was initially presented (abstract and results section) as being 15uN/min and then chosen in the middle of the results section to be adequate at 30uN/min, for reasons of inconsistency at the lower rate, but then in the conclusion section again 15uN/min is presented as the optimal one.
ii. The methodology section displays a formula for converting the Hertz modulus to Young’s Modulus, yet there are no Young’s moduli results reported at all in this manuscript so why is this formula needed?
iii. The introduction and abstract as well as the discussion suggest that this is a critical step towards creating a phantom that would enable better surgical training (I guess in accordance with the journal scope) but in fact they did nothing of that sort, apart from very basic mechanical characterization. They did not even suggest a strategy for implementing this data for the creation of more realistic phantoms. If this aim is only a very far and currently theoretical goal, and not yet shown in their current work, it would suffice to mention it once in the introduction while describing the impetus of this work and the overall rationale. Why give so much focus on it, if it is not relevant to the understanding of the results? for example, why do the readers need to know about the major causes of surgical failure and the most abundant complications? It would be better to emphasize instead the immediate contribution of this manuscript over the current literature, what knowledge gap does this data fill? what is novel or new about their findings?
iv. Others have previously characterized the pancreas rheological properties, but the authors claimed in the discussion that those prior studies did not perform those characteristics in a punctiform manner. Then even when a group did perform such punctiform measurements (Sugimoto et al, is mentioned by the authors, but for some reason the authors do not refer in the discussion to Rubiano et al, 2018) they did not evaluate the main pancreatic duct (MPD). However, also the authors did not show data of the MPD but only of the parenchyma so where is their contribution here? What is the major difference between their characterization of native tissue vs. that performed by others? Is it simply the understanding that freezing alters the mechanical properties? Or the fact that multiple locations are needed for this methodology?
Also, In the results section, the subtitle ‘Homogeneity’ appears twice once on page 10 and once on page 11, please amend. Most importantly, the first appearance on page 10 refers only to parenchymal homogeneity. The second appearance refers to freezing vs. fresh sample comparison. Is there any new data on the comparison between ductal vs. parenchyma characteristics?
b. There are many Figures and tables which makes it difficult to obtain a coherent and clear message. Presenting the raw data of individual measurement points and samples in each location is not very informative. Given the multitude of locations and samples, it may be better suited to create histograms of Hertz/Young’s moduli distribution in different sample types (e.g., parenchyma vs, duct or fresh vs. frozen etc.) instead of reporting the individual sample multipoint variability in terms of each point average and standard deviation. Reporting histograms that combine all the results obtained to better reflect or represent the overall spectrum of profiles that can be encountered in a single type of sample may be more informative. This is similar to what is usually shown in AFM mapping of surface properties, such as for instance performed in PMID: 29500447.
c. Figures level and resolution are not up to standards and in all cases lack statistical significance testing indication. For instance, the raw data images on Fig. 3 are not clear and not according to scientific reporting standards or even sizes, with rectangular vs. squarish graphs, lacking axis titles, unclear axis values and overall poor resolution. Please improve all figures of this manuscript. In addition, many figures have a title embedded in the graph itself, which is not common in scientific graph presentations (the figure caption already contains the title). There is no statistical testing performed or indicated in all figures. Also, Figure 4 and Table 1 show the same data, please choose one format for the main text, and put the other in the supplementary material. The same for figures 5-8 and tables 2-5, and figures 10-11 and tables 6-7, respectively (both in terms of statistical testing results indication and redundancy reduction). Similar remarks are also related to all other figures – please improve them.
d. English level is a bit wordy, and on many cases could be shortened to increase clarity. I suggest English editing for this manuscript by a native English Speaker.
e. Overall, the level of mechanical engineering science in this study is basic. The methods section is more like a protocol handbook rather than a professional methodology description of the experiments done. While the definition of variables and terminology is commendable, it should be better written, and not in bullet points.
f. The authors characterized the elasticity modulus at low indentation rates, assuming the behavior at that rate is almost static. However, soft tissues, and pancreas is no exception, have a viscoelastic behavior, as the authors themselves acknowledge in some parts of this manuscript, yet they did nothing to characterize it. It would be much improved if the authors could add to it important mechanical and modeled data which include, but is not limited to, measuring hysteresis, and energy dissipation during single vs. repeated cycles, using other models (such as maxwel, Kelvin-Voigt, and Zener models) to characterize the relative contributions of elasticity (‘spring’) and viscosity (‘dashpot’) components (some of which was previously reported by the same Rubiano et al. Acta Biomaterialia, 2018 paper) to the pancreas mechanical properties in relation to the Hertz modulus, and/or provide finite element modeling of the pancreas mechanical distribution based on the empirical results obtained.
g. Please clarify in the methodology section whether any pre-conditioning was performed to ensure stable and representative values are measured in each indentation, what was the time interval between the repeated measures (for example in the case of determination of optimal indentation rate to assure that one measurement did not affect the consequent repeated measurements in that same location), and show either in the results or the supplementary information supportive figure showing the results of repeated measurements in the same location to allow estimation of the pre-conditioning required prior to actual measurements that reflect ‘real’ values, in each measurement rate tested.
h. Also please better explain the rationale of the testing profile, why was a pause made and based on what was the duration of 10 min chosen as sufficient?
i. In cases of multipoint measurements, please explain how many points were measured for each sample, at what distance apart, what was the rationale behind the choice of points (random or according to a certain scheme), and was there any measurement of specific duct locations, or if not how were these avoided?
2. Minor comments:
a. Fig.1 caption: the word “smother” should probably be “smoother”
b. Formulas should be numbered, and references made clear as to their source.
c. Please state the sample number, number of points, and number of measurements performed in each point clearly for each measurement performed (rate determination, single vs. multipoint, before and after freezing etc.)
d. Please make sure all formulas and variables are clearly defined – for instance, it is not clear what the ‘reduced E’ (or Er) stands for.
Comments on the Quality of English LanguageWould benefit from professional English editing.
Author Response
Dear Reviewer,
Thank you for your review of our paper. We have answered each of your points below.
- Major comments:
- Overall the work presented is not sufficiently coherent. To name but a few examples:
- The optimal rate was initially presented (abstract and results section) as being 15uN/min and then chosen in the middle of the results section to be adequate at 30uN/min, for reasons of inconsistency at the lower rate, but then in the conclusion section again 15uN/min is presented as the optimal one.
Standard BS ISO 37:2017 suggest the use of low velocity to test viscoelastic material, but there are no Standard specifically written for the biological materials described in or work. This is why we started from the velocity suggested by the manufacture of the instrumentation used (Anton Paar) and we decrease it.
As it is generally known from the theory on the characterization of viscoelastic material, the lower the speed of execution of the tests, the more plausible the result due to the reduction of the dissipated energy (typical for these kind of material) when a load is applied, then removed.
We started by reducing the loading rate from 180 uN/min to 30 uN/min observing large variations (statistically different from each other) in the elastic modulus values obtained. Then, we moved to 15 uN/min observing a slight decrease in the elastic modulus compared to 30 uN/min, but without a statistically significant difference. Considering the lower loading rate, we risked going close to the full-scale value of the machine in terms of displacement, so we therefore decided to use 30 uN/min loading rate.
However, thanks to your comment, we noticed the inconsistency in reporting this information and changed it in the different sections of the paper (p. 1, 6, 16)
- The methodology section displays a formula for converting the Hertz modulus to Young’s Modulus, yet there are no Young’s moduli results reported at all in this manuscript so why is this formula needed?
When talking about E we referred to the elastic modulus since, for indentation with a spherical probe, it can be approximated with Hertz’s modulus.
However, the equations have been updated in the hopes of increasing the clarity of the manuscript. The Hertz modulus is the only estimation of elastic modulus of the material used in this paper, and is therefore generally referred as elastic modulus in some of the following paragraphs.
- The introduction and abstract as well as the discussion suggest that this is a critical step towards creating a phantom that would enable better surgical training (I guess in accordance with the journal scope) but in fact they did nothing of that sort, apart from very basic mechanical characterization. They did not even suggest a strategy for implementing this data for the creation of more realistic phantoms. If this aim is only a very far and currently theoretical goal, and not yet shown in their current work, it would suffice to mention it once in the introduction while describing the impetus of this work and the overall rationale. Why give so much focus on it, if it is not relevant to the understanding of the results? for example, why do the readers need to know about the major causes of surgical failure and the most abundant complications? It would be better to emphasize instead the immediate contribution of this manuscript over the current literature, what knowledge gap does this data fill? what is novel or new about their findings?
Yes, the Reviewer is right. The creation of the pancreatic phantom is a critical step according to the journal scope. The learning curve for the pancreatic surgery is very slow [Hardacre, J. M. Is there a learning curve for pancreaticoduodenectomy after fellowship training? HPB Surgery 2010, 6–9; Fisher, W.E.; Hodges, S.E.; Wu, M.F.; Hilsenbeck, S.G.; Brunicardi, F.C. Assessment of the learning curve for pancreaticoduodenectomy. American Journal of Surgery 2012, 203, 684–690; Tsamalaidze, L.; Stauffer, J. A. Pancreaticoduodenectomy: minimizing the learning curve. Journal of Visualized Surgery 2018, 4, 64–64], and due to the differences in the mechanical properties between human and cadaver pancreatic tissue, the best way to reduce the gap is to have surgeons practice on a phantom that fully replicates the mechanical characteristics of the pancreas itself.
Even if the production of the phantom is distant from this paper, our work can lead to several more immediate applications. One of them is the analysis of the precision of the surgeon's evaluation the importance of which has already been highlighted by Sugimoto et all. We have been collected data from the clinician and will publish on this issue as soon as the numerosity is acceptable. Analogously a comparison with the maps obtained from elastography is being developed.
Pancreatic tissue is very challenging to characterize, including the impossibility of creating planar sections with constant thickness, as is customary performing indentation of rigid material. So, a preliminary characterization could be an interesting starting point for the selection of the proper material in order to obtain a synthetic phantom. According to the Reviewer comment, we add a sentence that better describe the implementation of the strategy to obtain the phantom (“The modulus here estimated will be the first selection criteria for the artificial materials. Only for those materials that will show the same indentation modulus the viscoelastic characteristics will be analysed. Once the material is selected, a 3D model of the organ will be recreated from clinical images that will be appropriately modified by removing tissues that will not match with the region of interest. Using 3D printing or cast techniques (depending on the material selected), the phantom will be obtained.”, p.14). Moreover, we clarified in the abstract as the preliminary characterization of the physiological pancreatic tissue is the objective of this work, with the long-term goal of obtaining a faithful synthetic replica of the human pancreas. (“The aim of this study was to analyze the preliminary mechanical characteristics of the pancreas to create a realistic synthetic phantom for surgical simulations in the near future.”). We have also modified the paper, mentioning it once in the introduction and in the conclusion.
Even though the nature of the test might appear basic, there is a clear lack of a precise and reproducible protocol in the literature, resulting in data for which a meaningful comparison is challenging. Sugimoto and colleagues, for example, used a large 5 mm point on a vaguely described instrument while the cantilever-based instrument used by Rubiano is custom and therefore hard to reproduce. Even if Rubiano and colleagues were very clever in being able to get flat, parallel surfaces slicing the sample in a stainless-steel matrix slicer, they didn’t fix the sample in their custom-made holder where the samples could be able to have even small movements affecting the test results. Moreover, they probably did not use an anti-vibration platform, which is essential when the characterization forces involved in the test are on the order of tens of uN. In addition, Sugimoto and colleagues repeated the test 3 times on each point without pauses probably altering the average results obtained. Our analysis has been performed using a commercial instrument and, with the parameters described, it should be simple for other research groups to replicate the results. Additionally, to our knowledge, no other group has presented results on multipoint analysis, freezing effects and MPD characterization (that we have add, see next comment).
- Others have previously characterized the pancreas rheological properties, but the authors claimed in the discussion that those prior studies did not perform those characteristics in a punctiform manner. Then even when a group did perform such punctiform measurements (Sugimoto et al, is mentioned by the authors, but for some reason the authors do not refer in the discussion to Rubiano et al, 2018) they did not evaluate the main pancreatic duct (MPD). However, also the authors did not show data of the MPD but only of the parenchyma so where is their contribution here? What is the major difference between their characterization of native tissue vs. that performed by others? Is it simply the understanding that freezing alters the mechanical properties? Or the fact that multiple locations are needed for this methodology?
Thank you for your comment. The data on MPD characterization was initially excluded from this work due to a very low numerosity. In fact our samples come from the resection of healthy tissue from a pathological pancreas and the duct was not always available due to surgical reasons. Since in the last weeks we have been able to collect more samples and the available data has increased significantly, we have decided to include them in this work (p.10-11).
In addition, we believe that information on the mechanical behaviour of the pancreatic tissue according to the type of preservation (fresh VS frozen), as well as the loading rate used, could be an interesting contribution to the paucity of literature on the pancreatic tissue characterization compared to that available for other tissues (organs or parts thereof).
- Also, In the results section, the subtitle ‘Homogeneity’ appears twice once on page 10 and once on page 11, please amend. Most importantly, the first appearance on page 10 refers only to parenchymal homogeneity. The second appearance refers to freezing vs. fresh sample comparison. Is there any new data on the comparison between ductal vs. parenchyma characteristics?
We apologise as that was a mistake on our part, the title of the section at page 11 has been changed to “Freezing”.
- There are many Figures and tables which makes it difficult to obtain a coherent and clear message. Presenting the raw data of individual measurement points and samples in each location is not very informative. Given the multitude of locations and samples, it may be better suited to create histograms of Hertz/Young’s moduli distribution in different sample types (e.g., parenchyma vs, duct or fresh vs. frozen etc.) instead of reporting the individual sample multipoint variability in terms of each point average and standard deviation. Reporting histograms that combine all the results obtained to better reflect or represent the overall spectrum of profiles that can be encountered in a single type of sample may be more informative. This is similar to what is usually shown in AFM mapping of surface properties, such as for instance performed in PMID: 29500447.
Yes, the reviewer is right. In order to improve the readability of the manuscript we remove several Figures (5,6,7), while Tables 1 through 5 are replaced by a single table (new Table 2).
We appreciate a lot the suggestion on the representation as usually show in AFM mapping or using histogram of the elastic modulus made by Hopfa and Pierce (doi: 10.1016/j.mspro.2014.10.028), but we believe that this graphical representation is more suitable for a large amount of data."
- Figures level and resolution are not up to standards and in all cases lack statistical significance testing indication. For instance, the raw data images on Fig. 3 are not clear and not according to scientific reporting standards or even sizes, with rectangular vs. squarish graphs, lacking axis titles, unclear axis values and overall poor resolution. Please improve all figures of this manuscript. In addition, many figures have a title embedded in the graph itself, which is not common in scientific graph presentations (the figure caption already contains the title). There is no statistical testing performed or indicated in all figures. Also, Figure 4 and Table 1 show the same data, please choose one format for the main text, and put the other in the supplementary material. The same for figures 5-8 and tables 2-5, and figures 10-11 and tables 6-7, respectively (both in terms of statistical testing results indication and redundancy reduction). Similar remarks are also related to all other figures – please improve them.
Yes, the reviewer is right. As already mentioned in the last comment, in order to improve the readability of the manuscript we remove several Figures (5,6,7), while Tables 1 through 5 are replaced by a single table (new Table 2). All the resolutions of the remaining graphics have been improved.
- English level is a bit wordy, and on many cases could be shortened to increase clarity. I suggest English editing for this manuscript by a native English Speaker.
Thank you for your comment, the manuscript has been revised. Anyway, the first Author has international English certifications (CPE, grade B – OET, grade B).
- Overall, the level of mechanical engineering science in this study is basic. The methods section is more like a protocol handbook rather than a professional methodology description of the experiments done. While the definition of variables and terminology is commendable, it should be better written, and not in bullet points.
Thank you for your observation, several portions of the paper have been revised with this in mind. We tried to write this paper in accordance with the more clinical aspect that the journal is focused on. Following data will likely be published in a more engineering focused journal where methodology and technical considerations will be given higher relevance.
- The authors characterized the elasticity modulus at low indentation rates, assuming the behavior at that rate is almost static. However, soft tissues, and pancreas is no exception, have a viscoelastic behavior, as the authors themselves acknowledge in some parts of this manuscript, yet they did nothing to characterize it. It would be much improved if the authors could add to it important mechanical and modeled data which include, but is not limited to, measuring hysteresis, and energy dissipation during single vs. repeated cycles, using other models (such as maxwel, Kelvin-Voigt, and Zener models) to characterize the relative contributions of elasticity (‘spring’) and viscosity (‘dashpot’) components (some of which was previously reported by the same Rubiano et al. Acta Biomaterialia, 2018 paper) to the pancreas mechanical properties in relation to the Hertz modulus, and/or provide finite element modeling of the pancreas mechanical distribution based on the empirical results obtained.
Thank you for your comment. We are in fact planning to perform a full viscoelastic characterization including creep, relaxation, hysteresis, energy dissipation and DMA. We are waiting on these results to perform any type of modelling as to have a more appropriate characterization. As before mentioned, it is our intention to publish these following results on a more engineering focused journal, in order that they may be more appreciated.
- Please clarify in the methodology section whether any pre-conditioning was performed to ensure stable and representative values are measured in each indentation, what was the time interval between the repeated measures (for example in the case of determination of optimal indentation rate to assure that one measurement did not affect the consequent repeated measurements in that same location), and show either in the results or the supplementary information supportive figure showing the results of repeated measurements in the same location to allow estimation of the pre-conditioning required prior to actual measurements that reflect ‘real’ values, in each measurement rate tested.
No preconditioning was performed on the samples. Cyclic tests are planned in order to analyse whether or not it will be necessary for DMA testing. The 10 minutes pause was found to be appropriate after test were performed with 2, 5 and 10 minutes. Since these results were only preliminary, they have not been included also due to the more clinical nature of the journal.
- Also please better explain the rationale of the testing profile, why was a pause made and based on what was the duration of 10 min chosen as sufficient?
Thank you for your question which we hoped helped us better clarify the nature of our protocol. The following section was added on pg 5:
“For each point 5 indentations were performed for statistical significance at 10 minutes intervals. The pause between indentation was deemed necessary since no cyclic characterization of the pancreatic tissue was available and in was therefore not possible to predict the effect of multiple consecutive indentations. 10 minutes proved to be enough since no trend (either hardening nor softening) was evident in our tests.”
- In cases of multipoint measurements, please explain how many points were measured for each sample, at what distance apart, what was the rationale behind the choice of points (random or according to a certain scheme), and was there any measurement of specific duct locations, or if not how were these avoided?
Following your suggestion, the following paragraph was added on pg 5 in the hopes of increasing the clarity on our protocol:
“For each sample multiple points (3 to 5) were tested according to the size of the specimen. The points where distributed as evenly as possible across the sample; in any case a distance of at least 2 mm was kept in order to ensure no overlapping of the indentation areas. Points with irregular surfaces (peaks and valleys) were avoided as to have a clean indentation curve.”
Moreover, duct identification is now explained on pg 2:
“The MPD was extracted from the sample and cut longitudinally to allow the indentation of the inner surface of the duct.”
- Minor comments:
- Fig.1 caption: the word “smother” should probably be “smoother”
We apologise as this was a mistake on our part, the caption has been changed accordingly.
- Formulas should be numbered, and references made clear as to their source.
We apologise as this was a mistake on our part, the caption has been added accordingly.
- Please state the sample number, number of points, and number of measurements performed in each point clearly for each measurement performed (rate determination, single vs. multipoint, before and after freezing etc.)
Thank you for your comment. A table has been added on pg 5 with that information.
- Please make sure all formulas and variables are clearly defined – for instance, it is not clear what the ‘reduced E’ (or Er) stands for.
The equations were rewritten in order to improve readability of the manuscript, and a definition of the variable was included:
“The reduced modulus Er [Pa], a fictitious value in which the elastic deformation of the sample and the indenter are combined, is estimated from the indentation”.

Reviewer 3 Report
Comments and Suggestions for Authors
Overall comments:
The subject matter of this manuscript deals with the analysis of the mechanical characteristics of the pancreas through nano-bio-indentation. The authors showed that a slow loading rate, immediate analysis, and multipoint measurements are essential for obtaining accurate values of the elastic modulus. The results emphasize the importance of these parameters in creating a realistic synthetic phantom for surgical simulations and caution against analyzing specimens post-freezing, as it may alter the outcomes.
I’d like to recommend that the authors try to answer the following questions in their revised version, which in my opinion, will significantly improve the possible attractiveness of this work for citation. Furthermore, if such issues are all cleared by the authors, this paper seems to be qualified to secure its publication.
Specific concerns:
1. In Figure 3, please provide a legend for each line increase the size of the numbers on the X and Y axes to improve readability. Also, make clear the graph line to distinguish between force and displacement in 3(B).
2. In Figure 8, why is the standard deviation for Sample 2 is too large? Reproducibility is unreliable due to the large standard deviation.
3. Overall, uniform the font for all graphs in Figures including labels of X and Y axes, legends, and units.
4. Why does Loading rates for 15 µN/min of 4A point not measure?
5. What was confirmed through the force-displacement curve in Figure 9 and the average value of EHz in Figure 10, and what does it mean? It is necessary to discuss and analyze about the result.
6. The author said that “there is a statistically significant difference in the medium EHz of the pancreatic parenchyma before (2.12 ±0.90 kPa) and after (1.86 ±0.87 kPa) the freezing process”. Please note asterisks (*p < 0.05, **p < 0.01, ***p < 0.001, and ****p < 0.0001) between Before freezing group and After freezing group in Figure 11 to support the mentioned results.
Comments on the Quality of English LanguageMinor editing of English language required.
Author Response
Dear Reviewer,
Thank you for your review of our paper. We have answered each of your points below.
- In Figure 3, please provide a legend for each line increase the size of the numbers on the X and Y axes to improve readability. Also, make clear the graph line to distinguish between force and displacement in 3(B).
Thank you for your comment, the figure has been updated in the hopes of increasing its clarity.
- In Figure 8, why is the standard deviation for Sample 2 is too large? Reproducibility is unreliable due to the large standard deviation.
The samples are often heterogenous, leading to large standard deviation. In particular for sample 2 the position 2A was significantly harder than 2B and 2C as shown in figure 5.
- Overall, uniform the font for all graphs in Figures including labels of X and Y axes, legends, and units.
Following your suggestion all the figures of the manuscript have been updated.
- Why does Loading rates for 15 µN/min of 4A point not measure?
We appreciate the Reviewer's observation. Due to certain machine limitations, some of the indentations at the slowest rate were unsuccessful. The lower the elastic modulus of the samples, the higher the displacement of the indenter tip. When reducing the loading rate, we sometimes observe a decrease in the elastic modulus, resulting in an increase in displacement to achieve the same test force. Specifically, for that point, it was not possible to perform the analysis. This issue has been further discussed on page 9:
“With the extra slow loading (15 µN/min) however, some of the indentations were not performed correctly as the indentation curve had more artefacts and the displacement range was at times saturated.”
- What was confirmed through the force-displacement curve in Figure 9 and the average value of EHz in Figure 10, and what does it mean? It is necessary to discuss and analyze about the result.
Thank you for your comment. Following your suggestion this paragraph was added on pg 11 to comment on these results:
“According to the statistical test performed the 5 points are not equivalent. In particular point A results statistically significantly different from all others except D and many other differences arise between the points. The comparison between each point is shown in table 6 in the appendix. Considering the statistically significant differences that can be seen within the same sample it was concluded that a minimum of three points had to be analyzed for each sample in order to get a correct characterization.”
The mentioned table can be found in the added appendix on pg 17.
- The author said that “there is a statistically significant difference in the medium EHz of the pancreatic parenchyma before (2.12 ±0.90 kPa) and after (1.86 ±0.87 kPa) the freezing process”. Please note asterisks (*p < 0.05, **p < 0.01, ***p < 0.001, and ****p < 0.0001) between Before freezing group and After freezing group in Figure 11 to support the mentioned results.
Thank you for your comment. Figure 11 has been updated to include the mentioned notation.

Round 2
Reviewer 2 Report
Comments and Suggestions for Authors
I would like to thank the authors for the thorough revision, which addressed most of my concerns. I have two comments that still require the authors' attention:
1. The English level could still be improved to make the message clearer to a wide audience. I strongly suggest this manuscript be reviewed by an expert and native English speaker to increase legibility and clarity.
2. The authors have made significant changes to most figures bringing them to an acceptable level. Figure 4, though, still shows inadequately presented data. This figure presumably shows raw output data from the indentation machine. However:
a) The graphs' axes are not legible
b) The font sizes are too small, and their resolution is not sufficient to be legible in digital zoom. For instance, the point determination (blue) values in panel A are not legible nor clear in the graph itself where you can only see green and red lines.
(c) The axes, in many cases, lack axes titles.
(d) The arrangement of the different graphs side by side (panels A and B) is not symmetrical in size, and is non-aesthetic,
(e) The top figure in panel B overlaps two lower figures to the extent that it hides valuable data.
Please rectify this figure accordingly.
Comments on the Quality of English Language
Kindly see my main point above - please have the manuscript reviewed by a professional native English speaker to increase clarity.
Author Response
We thank the Reviewer for this comment. We have answered each of your points below.
I would like to thank the authors for the thorough revision, which addressed most of my concerns. I have two comments that still require the authors' attention:
- The English level could still be improved to make the message clearer to a wide audience. I strongly suggest this manuscript be reviewed by an expert and native English speaker to increase legibility and clarity.
Thank you for your comment, English has been revised and ameliorated. In this context, we underline that the first author (Dr. Pagnanelli) passed various English Certificates as CPE (grade B) and OET medicine (grade B).
- The authors have made significant changes to most figures bringing them to an acceptable level. Figure 4, though, still shows inadequately presented data. This figure presumably shows raw output data from the indentation machine. However:
We are really sorry but we believe that there was an error in the layout of the manuscript in the creation of the PDF of the paper. Below the Reviewer can see the image that we had uploaded as Figure 4,
While this is what you see by downloading the paper in PDF format (see PDF)
We are sorry for the inconvenience and understand all the reviewer's concerns.
However, since some comments could have been directed to our modified version of Figure 4 as well, we decided to make some changes as shown in the new image.
- a) The graphs' axes are not legible
Thank you very much for your comment: we increased the font size and improving graphic quality
- b) The font sizes are too small, and their resolution is not sufficient to be legible in digital zoom. For instance, the point determination (blue) values in panel A are not legible nor clear in the graph itself where you can only see green and red lines.
The new image should make the meaning of the green, red and blue lines clearer.
(c) The axes, in many cases, lack axes titles.
The panel without the axes were already removed in the past version of the Figure 4.
(d) The arrangement of the different graphs side by side (panels A and B) is not symmetrical in size, and is non-aesthetic,
Thank you very much for your comment, now the two panels (A and B) were made symmetrical in size and position.
(e) The top figure in panel B overlaps two lower figures to the extent that it hides valuable data.
We are sorry but we this overlapping was due to the error in the layout of the manuscript in the creation of the PDF of the paper.
Please rectify this figure accordingly.
Thank you very much for your patience and for all the helpful comments that we hope to have implemented to the best of our ability.

Reviewer 3 Report
Comments and Suggestions for Authors
Authors have well changed their previous manuscript according to my suggestions and others’ ones. I have no further comments.
Author Response
Thank you for your revision and your helpful comments.
Best regards,
The Authors